

# Assessment of locomotive syndrome among older individuals: a confirmatory factor analysis of the 25-question Geriatric Locomotive Function Scale

Chaochen Wang[1], Tatsunori Ikemoto[2], Atsuhiko Hirasawa[2], Young-Chang Arai[3], Shogo Kikuchi[1] and Masataka Deie[2]

[1] Department of Public Health, Aichi Medical University, Nagakute, Aichi, Japan
[2] Department of Orthopedic Surgery, Aichi Medical University, Nagakute, Aichi, Japan
[3] Institute of Physical Fitness, Sports Medicine and Rehabilitation, Aichi Medical University, Nagakute, Aichi, Japan

Corresponding author
Chaochen Wang,
chaochen@wangcc.me

## ABSTRACT

**Background:** The 25-question Geriatric Locomotive Function Scale (GLFS-25) is widely used in daily clinical practice in evaluating locomotive syndrome (LS). The questionnaire contains 25 questions aiming to describe 6 aspects, including body pain, movement-related difficulty, usual care, social activities, cognitive status, and daily activities. However, its potential underlying latent factor structure of the questionnaire has not been fully examined so far.

**Methods:** Five hundred participants who were 60 years or older and were able to walk independently with or without a cane but had complaints of musculoskeletal disorders were recruited face to face at the out-patient ward of Aichi Medical University Hospital between April 2018 and June 2019. All participants completed the GLFS-25. Confirmatory factor analysis (CFA) models (single-factor model, 6-factor model as designed by the developers of the GLFS-25) were fitted and compared using *Mplus* 8.3 with a maximum likelihood minimization function. Modification indices, standardized expected parameter change were used, a standard strategy for scale development was followed in the search for an alternative and simpler model that could well fit the collected data. Cronbach's α and its 95% confidence interval (CI) were also calculated.

**Results:** Mean (standard deviation) participants age was 72.6 (7.4) years old; 63.6% of them were women. Under the current criteria, 132 (26.4%) and 262 (52.4%) of the study subjects would be classified as LS stage 1 and stage 2, respectively. Overall, the Cronbach's α (95% CI) for GLFS-25 evaluated using these data was 0.959 (0.953, 0.964). The single- and 6-factor models were rejected due to poor fit. The alternative models with either full 25 questions or a shortened GLFS-16 were found to fit the data better. These alternative models included three latent factors (body pain, movement-related difficulty, and psycho-social complication) and allowed for cross-loading and residual correlations.

**Discussion:** The findings of the CFA models provided evidence that the factor structure of the GLFS-25 might be simpler than the 6-factor model as suggested by the designers. The complex relationships between the latent factors and the observed items may also indicate that individual sub-scale use or simply

combining the raw scores for evaluation is likely to be inadequate or unsatisfactory. Thus, future revisions of the scoring algorithm or questions of the GLFS-25 may be required.

# INTRODUCTION

Locomotive syndrome (LS) was described by the Japanese Orthopaedic Association (JOA) in 2007 as a wide range of disabilities attributed to deterioration of the musculoskeletal components (*Nakamura, 2008*; *Nakamura & Ogata, 2016*). This concept has become widespread worldwide (*Yi & Lee, 2018*; *Huang et al., 2018*; *Tavares & Santos, 2017*). Although the original concept of LS included musculoskeletal disorders such as osteoarthritis, spondylosis and osteoporosis, recent studies have indicated that LS involves psychological disorders as well (*Ikemoto et al., 2016*; *Nakamura et al., 2017a*). Previous studies also indicated that LS, frailty and sarcopenia may contribute similarly in terms of the physical aspect of functional decline among older individuals (*Ikemoto & Arai, 2018*; *Yoshimura et al., 2019*).

In 2012, the JOA developed a screening tool for LS, the 25-question Geriatric Locomotive Function Scale (GLFS-25) (*Seichi et al., 2012*), a self-reported comprehensive measure that consists of 25 questions that refer to experiences in the preceding month. The criteria for identifying LS using GLFS-25 are formally 7~15 as stage 1 and 16~100 as stage 2 for the total score (range: 0–100 points) (*Nakamura & Ogata, 2016*). The scale was designed to address six aspects using 25 questions: four questions regarding body pain, three questions regarding movement-related difficulty, 5 questions regarding usual care, four questions regarding social activities, two questions regarding cognitive status, and seven questions regarding daily activities closely related with the five above-mentioned aspects.

The GLFS-25 had low response rate due to missingness when it was self-administered in previous studies, which may caused by conditions of LS among patients and/or the length of the questionnaire might be considered as too long for some individuals (*Ishii et al., 2015*). Previously, *Tanabe et al. (2018)* provided a shorter version of GLFS-25 through factor analysis. The shorter version of GLFS-25 included nine questions with three questions loaded onto each of three latent factors: outdoor activities, indoor activities, and lower limb functions. However, this shortened GLFS-25 is not widely accepted and considered unsuitable mainly because it ignored both the part of indicators for body pain, and the potential psychological aspect of visiting patients with LS. To date, no study has reported any adequate factor structure underneath the GLFS-25. It is also unclear whether the originally proposed 6-factor model provides the best fit to the collected data compared with the one- or other-structure factor models.

We aimed to examine the factor structure of the GLFS-25 in a sample of Japanese older individuals with musculoskeletal disorders. This is the first study to investigate the best fitting model of the factor structure of the GLFS-25 among a clinical sample with LS.

## METHODS

### Participants

Participants in this study ($n$ = 500) were recruited at the out-patient ward of Aichi Medical University Hospital between April 2018 and June 2019. The authors (T.I., A.H. & Y-C.A.) asked eligible subjects to answer the GLFS-25 questionnaire and collected questionnaires until 500 samples had been completed in their order of visiting. Questionnaires were obtained from over 95% of eligible subjects. Inclusion criteria were as follows: (1) age ≥60 years; (2) complaints of musculoskeletal disorders; and (3) ability to walk independently with or without a cane. Subjects with confirmed or suspected dementia were excluded from the study.

In estimating sample size, it is well understood that the use of larger samples in factor analyses tends to provide more precise estimates of population loading because as the number increases, sampling error is reduced; however, consensus is lacking about how large a sample is necessary to sufficiently achieve these aims. Previous literature reported that a sample of 500 or more is desirable in factor analysis (*MacCallum et al., 1999*).

We were able to achieve a 100% response rate from the 500 recruited subjects with the help of nurses and physicians who provided face-to-face interviews to comprehensively explain the purpose of the study as well as help them complete the questionnaire. For each participant, only the chief complaint was examined during the interview. Other chronic or musculoskeletal diseases which might also contribute to their answers on GLFS-25 were not checked by the physicians.

The study was approved by the Aichi Medical University Hospital Research Ethics Board (2019-H136). Requirement for written consent was waived for subjects evaluated during the study period unless they refused to provide the information in accordance with the withdrawal strategy.

### Model specification

According to the development of GLFS-25 (*Seichi et al., 2012*), a 6-factor model was specified to explore the latent structure of the questionnaire. Specifically, the six dimensions suggested were namely body pain, movement-related difficulty, usual care, daily activity, social activity, and cognition. The question items that were designed to be loaded on each of the above-mentioned latent factors are listed in Table 1 and illustrated in Fig. 1.

The proposed measurement model contained no cross-loading indicators (i.e., none were allowed to be loaded onto more than one latent variable), and all measurement errors were presumed uncorrelated. The first indicators for each latent variable (Q1, Q5, Q8, Q12, Q18, Q24) were used as marker indicators for the 6 latent factors. Accordingly, the model was over-identified with 260 degrees of freedom (df).

### Input data, model estimation, and potential model modification

Prior to the confirmatory factor anlaysis (CFA) analysis, descriptive statistics were performed of the raw participant response data and age (Table S1). Skewness and kurtosis of the variables were calculated and checked for normality. The sample

**Table 1 The 25-question Geriatric Locomotive Function Scale (GLFS-25), items aligned under the six latent factors as suggested by *Seichi et al. (2012)*.**

**Questionnaire items**

Body pain

1. Did you have any pain (including numbness) in your neck or upper limbs?

2. Did you have any pain in your back, lower back or buttocks?

3. Did you have any pain (including numbness) in your lower limbs?

4. To what extent has it been painful to move your body in daily life?

Movement-related difficulty

5. To what extent has it been difficult to get up from a bed or lie down?

6. To what extent has it been difficult to stand up from a chair?

7. To what extent has it been difficult to walk inside the house?

Usual care

8. To what extent has it been difficult to put on and take off shirts?

9. To what extent has it been difficult to put on and take off trousers and pants?

10. To what extent has it been difficult to use the toilet?

11. To what extent has it been difficult to wash your body in the bath?

12. To what extent has it been difficult to go up and down stairs?

Daily activity

13. To what extent has it been difficult to walk briskly?

14. To what extent has it been difficult to keep yourself neat?

15. How far can you keep walking without rest?

0 = 2–3 km; 1 = 1 km; 2 = 300 m; 3 = 100 m; 4 = 10 m

16. To what extent has it been difficult to go out to visit neighbors?

17. To what extent has it been difficult to carry objects weighing 2 kg?

18. To what extent has it been difficult to go out using public transportation?

19. To what extent have simple tasks and housework been difficult?

Social activity

20. To what extent have load-bearing tasks and housework been difficult?

21. To what extent has it been difficult to perform sports activities?

22. Have you been restricted from meeting your friends?

23. Have you been restricted from joining social activities?

Cognition

24. Have you ever felt anxious about falls in your house?

25. Have you ever felt anxious about being unable to walk in the future?

**Note:**

Except for Q15, the responses to the other 24 questions are: 0 (= no pain/difficulty/anxiety), 1 (= mild pain/difficulty/anxiety), 2 (= moderate pain/difficulty/anxiety), 3 (= considerably pain/difficulty/anxiety) and 4 (= extremely pain/difficulty/anxiety).

variance-covariance matrix (Table S2) was analyzed with *Mplus* 8.3 (*Muthén & Muthén, 2017*) using the MplusAutomation package (*Hallquist & Wiley, 2018*) in R (*R Core Team, 2019*), and a maximum likelihood minimization function was applied as suggested (*Finney & DiStefano, 2006*) if skewness less than 2 and kurtosis less than 7 were satisfied. Goodness of fit was evaluated using the standardized root mean square residual (SRMR), root mean square error of approximation (RMSEA) and its 90%
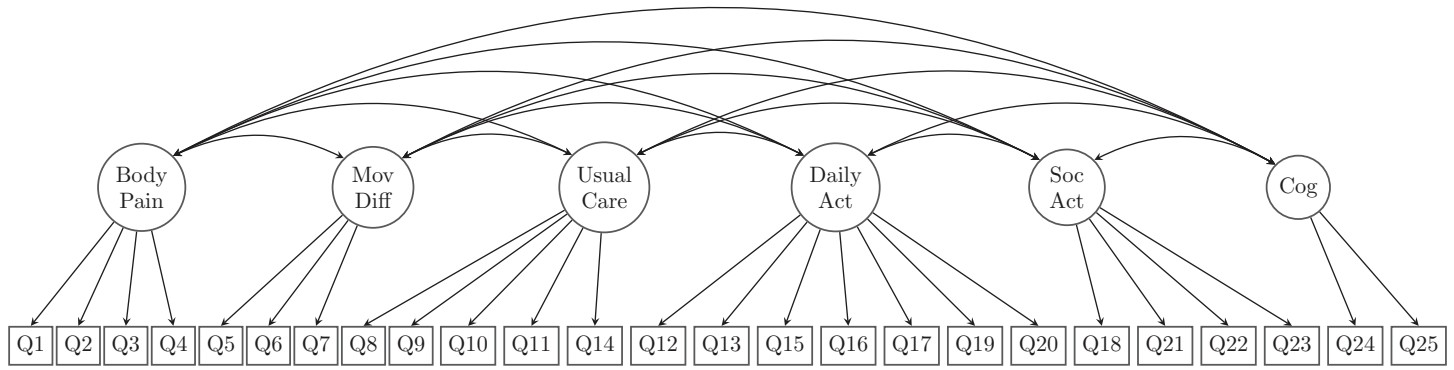

**Figure 1 Proposed structure of the 25-question Geriatric Locomotive Function Scale, GLFS-25 (the 6-factor model).** MovDiff, movement-related difficulty; DailyAct, Daily activity; SocAct, social activity; Cog, cognition.

confidence interval (CI), comparative fit index (CFI), and the Tucker–Lewis index (TLI). Guided by suggestions provided previously by *Brown (2014)*, and because different indices provide different information about model fit, acceptable model fit was chosen according to: RMSEA (≤0.08, 90% CI ≤0.08), SRMR (≤0.08), CFI (≥0.95), and TLI (≥0.95) (*Hu & Bentler, 1999*). Modification indices (MI), and standardized expected parameter change (SEPC) were used as guidance in model modification (*Brown, 2014*; *Whittaker, 2012*), and the final model was chosen based on the model fit criteria provided above. We followed the recommendation (*Brown, 2014*; *Whittaker, 2012*) by relaxing the fixed or constrained parameters from those with the largest modification index and standardized expected parameter change if this change (either cross-loading or error correlation) can be interpreted from empirical consideration. This process was repeated until we found that the model fit indices were mostly satisfied and potential re-specification would no longer improve model fit.

When latent factors were indicated to be highly correlated (factor correlation ≥0.85) (*Tabachnick, Fidell & Ullman, 2007*), the models were refitted with factors combined or loaded on to a second-order latent factor whichever the fit of the model can be better improved. For comparison, a CFA model with all observed variables loaded on a single factor was also fitted. This single-factor-model assumes that all the items were reflecting a homogeneous dimension of LS. Furthermore, we followed recommendation by *Worthington & Whittaker (2006)* in exploring a short form of GLFS. Specifically, *Worthington & Whittaker (2006)* suggested to examine factor loadings as a criterion for item retention. We retained items with loadings of 0.6 or greater on their latent factors, by which only the strongest indicators of the latent constructs of interest remain in the purified model.

Cronbach's α and its 95% confidence intervals (CI) were calculated for the collected data for comparison purpose with the developers' data.

# RESULTS AND DISCUSSION

## Results

Among the 500 participants, 63.6% (*n* = 318) were women. The mean age ± standard deviation of the whole sample was 72.6 ± 7.4 (range: 60–91 years). The median total

**Table 2 Summary of fit indices from the confirmatory factor analysis models for the GLFS-25.**

| Model | Single-factor model | 6-Factor model | Alternative model with cross loading and residual correlation allowed | Purified alternative model |
|---|---|---|---|---|
| $\chi^2$ | 2,287.699 | 1,332.957 | 780.385 | 215.858 |
| df | 275 | 260 | 250 | 90 |
| AIC | 27,736.751 | 26,812.009 | 26,279.438 | 16,689.283 |
| BIC | 28,052.847 | 27,191.324 | 26,700.898 | 16,950.589 |
| RMSEA | 0.121 | 0.091 | 0.065 | 0.053 |
| 90% CI | [0.116, 0.126] | [0.086, 0.096] | [0.060, 0.070] | [0.044, 0.062] |
| CFI | 0.797 | 0.892 | 0.947 | 0.978 |
| TLI | 0.779 | 0.875 | 0.936 | 0.971 |
| SRMR | 0.066 | 0.049 | 0.036 | 0.029 |

Note:
GLFS-25, the 25-question Geriatric Locomotive Function Scale; df, degree of freedom; AIC, Akaike information criterion; BIC, Bayesian information criterion; RMSEA, root mean square error of approximation; CI, confidence interval; CFI, comparative fit index; TLI, Tucker–Lewis index; SRMR, standardized root mean square residual.

GLFS-25 score was 16.9 (interquartile range, 7.7–32.8); 106 (21.2%) participants were not considered to have LS (GLFS-25 score <7), whereas 132 (26.4%) and 262 (52.4%) subjects met the criteria of LS stage 1 and stage 2, respectively, based on the latest definition of LS (*Nakamura & Ogata, 2016*). Cronbach's α for the GLFS-25 questionnaire in the present data was 0.959 (95% CI: [0.953, 0.964]) which was comparable with the GLFS-25 developers' data (*Seichi et al., 2012*). The majority of the responses were 0 or 1 to the question items. Less than 10% of the participants chose "considerably/extremely difficult" to questions regarding physical movement-related difficulty (Q5–Q10, Q14) and social- and family-related activities (Q16, Q19). Only 8% chose "considerably or extremely anxious" in response to the question that asked about their degree of anxiety about falling (Q24).

CFA model fit statistics are summarized in Table 2. The initial model tested was one in which each item was loaded on only one latent variable. Not surprisingly, this single-factor model did not fit the data well from either a statistical ($\chi^2$ = 2287.699, df = 275) or a practical (RMSEA = 0.121, 90% CI [0.116, 0.126]; CFI = 0.797; TLI = 0.779; SRMR = 0.066) perspective. The 6-factor model (Fig. 2) had somewhat improved model fit ($\chi^2$ = 1332.957, df = 260; RMSEA = 0.091, 90% CI [0.086, 0.096]; CFI = 0.892; TLI = 0.875; SRMR = 0.049) but were still unsatisfactory. Therefore, these models were rejected.

Due to the possibility of similar wordings in questions, residual correlations and cross-loadings between the latent variables were allowed in the modified alternative model based on both MI and SPEC. The modified alternative model indices (Table 2) confirmed that freeing the parameters of the cross loadings and residual correlations achieved substantial improvement ($\chi^2$ = 780.385, df = 250; RMSEA = 0.065, 90% CI [0.060, 0.070]; CFI = 0.947; TLI = 0.936; SRMR = 0.036). The model structure and its estimates of the loadings as well as the parameters are shown in Fig. 3. In the alternative model, we combined the latent variables of usual care together with movement-related difficulty (as MovDiff in Fig. 3) and social activity together with daily activity (as Soc-DailyAct in
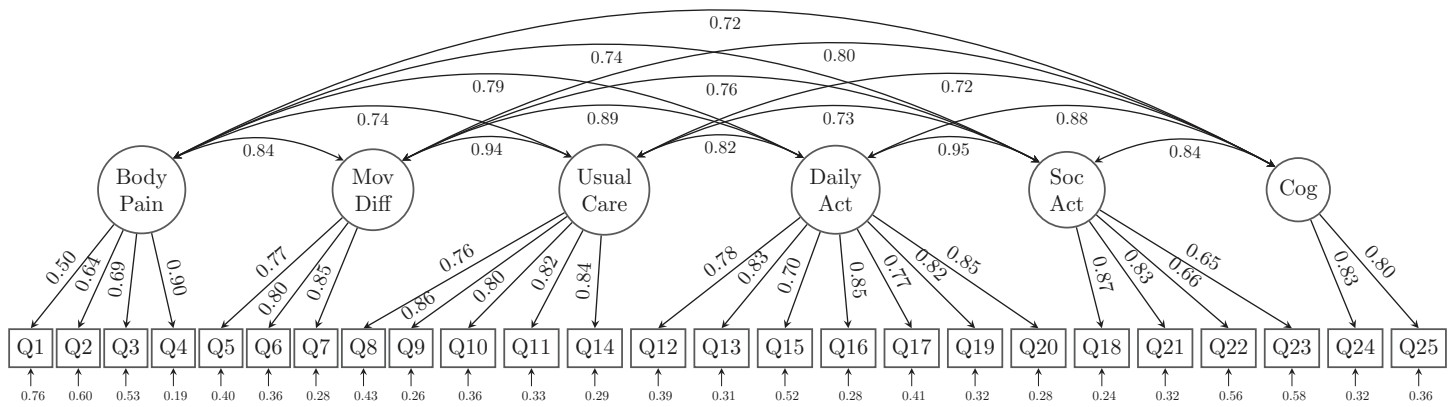

**Figure 2 The 6-factor model structure of the GLFS-25 with factor loadings.** MovDiff, movement-related difficulty; DailyAct, Daily activity; SocAct, social activity; Cog, cognition.

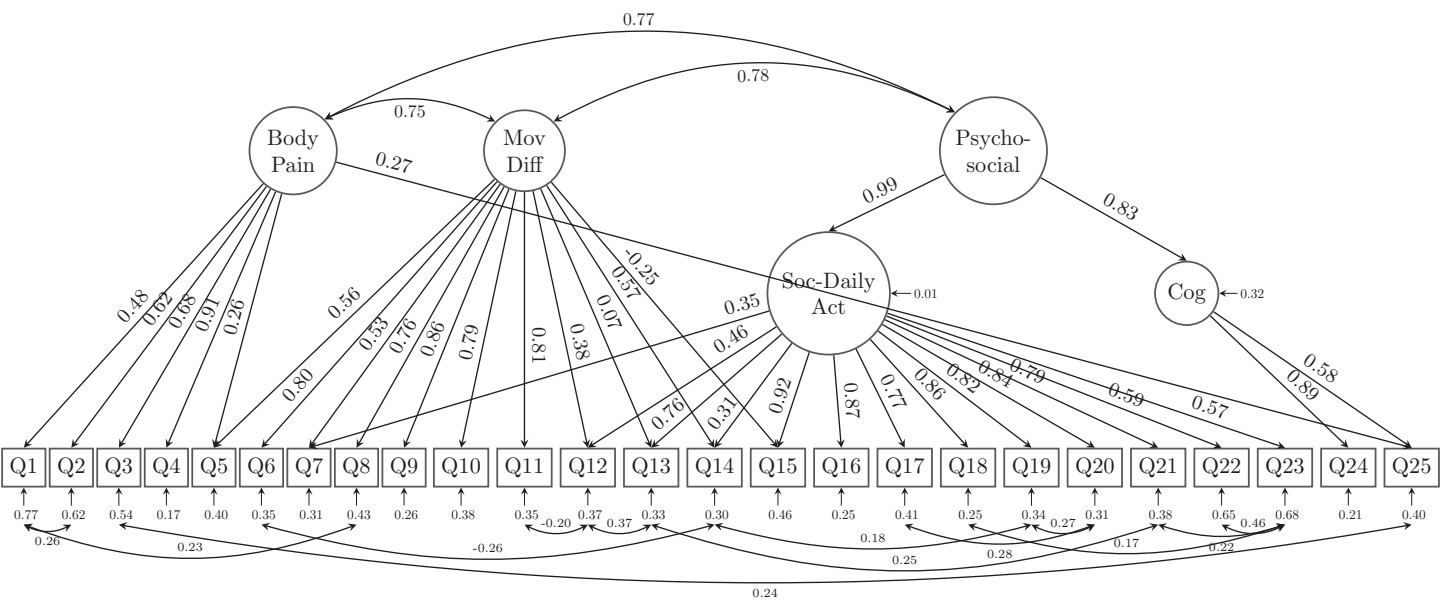

**Figure 3 The alternative model structure of the GLFS-25 with factor loadings.** MovDiff, movement-related difficulty; Soc-DailyAct, Social and Daily activity; Cog, cognition; Pycho-social, psycho-social complication.

Fig. 3) as they showed high intercorrelation (0.94 and 0.95, respectively) in the 6-factor model (Fig. 2). Furthermore, a second-order factor (Psycho-Social in Fig. 3) was found to be able to explain the factors of cognition (anxiety of falling) and Soc-DailyAct (social and daily activity). We also allowed cross-loading and residual correlations in the alternative model. Specifically, the factor body pain was positively associated with Q25 (anxiety of losing the ability to walk in the near future). Additionally, Q7, Q12, Q13, Q14 and Q15 are loaded on both factors of movement-related difficulty (MovDiff) and social daily activity (Soc-DailyAct); 13 pairs of residual correlations were allowed to

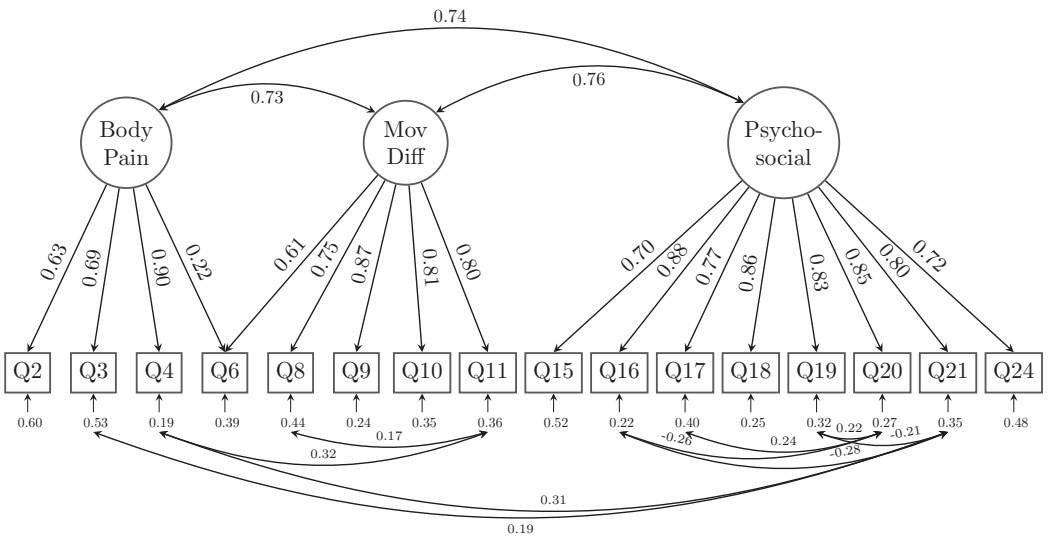

**Figure 4 The alternative model structure of the simplified GLFS-16 with factor loadings.** MovDiff, movement-related difficulty; Pycho-social, psycho-social complication.

achieve a better model fit on the collected data (i.e., Q1 with Q2; Q1 with Q8; Q6 with Q14; Q3 with Q25; Q11 with Q12; Q12 with Q13; Q13 with Q21; Q14 with Q19; Q17 with Q20; Q18 with Q22; Q19 with Q20; Q21 with Q23; and Q22 with Q23). Based on these analyses, the single-factor model and the proposed 6-factor model structure did not fit the data in the sample. An alternative model that reduced the number of latent factors from 6 to 3 (body pain, movement-related difficulty, and psycho-social complication) and allowed for cross-loadings and residual correlations showed significant improvement in model fit. All error variances were statistically significant ($p < 0.001$) in the presented alternative model.

However, as the CFI and TLI of the alternative model are still not satisfactory (both lower than 0.95), we scrutinized the loadings of the 25 indicators (questions) to retain those items with at least one loadings of 0.60 or greater on their latent factor. A total of 9 relatively lower-loading items were removed and resulted in a 16-question with 3 latent factor model (GLFS-16 as shown in Fig. 4). Specifically, Body pain retained 4 questions (Q2-Q4 and Q6), Movement-related difficulty retained 5 questions (Q6, Q8-Q11), whereas Psycho-social complication retained 8 questions (Q15–Q21, Q24). Except for Q6 which asked about the difficulty to stand up from a chair was cross-loaded onto both latent factors of body pain and movement-related difficulty (MovDiff). This shortened model provided acceptable data-model fit on all fit indices, $\chi^2 = 215.858$, df = 90; RMSEA = 0.053, 90%CI [0.044, 0.062]; CFI = 0.978, TLI = 0.971; SRMR = 0.029 (Table 2).

## DISCUSSION

We examined the factor structure of the GLFS-25 via CFA. The overall results of the model fit and its indices provided evidence supporting a 3-domain structure (body pain,

movement-related difficulty, psycho-social complication) of either the full 25 questions scale or a shortened GLFS with 16 questions. The shortened GLFS was suggested and found to best fit the data among the examined hypothesized models in a sample of Japanese older people with musculoskeletal disorders. While the original version of the GLFS-25 did not consist of distinct dimensions, our findings reduced the original 6 aspects (body pain, movement-related difficulty, usual care, daily activity, social activity, and cognition) down to 3 domains (body pain, movement-related difficulty, and psycho-social complication).

LS is based on a decline in motor functions that occurs with aging. It is important to recognize that LS is not just a pathological entity, but rather an epidemiological concept for healthcare system; thus, not only physical function tests (e.g., 2-step test, stand-up test) but also assessment using the GLFS-25 remain the official criteria for the diagnosis of LS (*Nakamura & Ogata, 2016*). Dimension and factor analysis are used to identify unique constructs. The GLFS-25 consists of several dimensions based on the current analysis, while the dimensions of the GLFS-25 were mostly overlooked in the assessment of LS in clinical practice. *Tanabe et al. (2018)* conducted an exploratory factor analysis of the GLFS-25 to develop a short version questionnaire. Their analysis reduced the 25 questions of the GLFS to 9 questions, 3 questions for each latent factor—daily outdoor living matters (Q18, Q22, Q23), indoor living matters (Q8, Q10, Q11), and lower limb function (Q3, Q12, Q13). However, the fact that this model ignored both pain in the body and psychological concerns has lowered its usefulness and representativeness. Not surprisingly, this model had poor model fit among our data and was therefore rejected ($\chi^2$ = 185.749, degree of freedom = 24; AIC = 10619.892, BIC = 10746.330; RMSEA = 0.116 (90% CI [0.101, 0.132]), CFI = 0.933, TLI = 0.900, SRMR = 0.038). On the other hand, our results showed three different domains (body pain, movement-related difficulty, and psycho-social complication) were possible latent factor structures behind the definition of LS. The conceptual structure of LS indicated that locomotive organ impairments would cause pain, stiffness, muscle weakness, and reduced balance function, leading to standing and walking difficulties followed by the need for long-term care (*Nakamura & Ogata, 2016*). Although our results did not theoretically discriminate among the 3 factors completely without cross-loading and residual correlations, it would potentially be able to assist clinical physicians in understanding the concept of LS (pain, movement-related difficulty and psycho-social complication) because these factors fundamentally contribute to the concept of LS and are closely linked with each other. Moreover, previous studies also reported relationships between musculoskeletal pain (*Hirano et al., 2014*; *Iizuka et al., 2015*; *Imagama et al., 2017*; *Kimachi et al., 2019*), movement-related difficulty or functional decline (*Akai et al., 2016*; *Iwaya et al., 2017*; *Izawa et al., 2019*; *Muramoto et al., 2012*; *Yoshimura et al., 2011*), and psycho-social complication (*Ikemoto et al., 2016*; *Nakamura et al., 2017a*, *2017b*; *Maruya et al., 2018*) and LS. Thus, it seems reasonable to discriminate among these 3 dimensions when assessing LS during clinical practice.

The alternative model with all 25 questions contained cross-loading of indicators to multiple latent factors. Methodologically, the cross-loading suggested that some questionnaire items are actually explained by more than one latent factor. For example, Q5 asked about difficulty getting up/lying down and was initially designed to reflect patients' movement difficulties. However, the revised alternative model suggested that the latent factor of body pain was also responsible for patients' answers to this question. Other questions such as Q7 (walking indoors), Q12 (using the stairs), Q13 (ability to walking briskly), Q14 (becoming presentable before going out), and Q15 (longest distance one can walk) were also found to be cross-loaded on both movement-related difficulty and social-daily activity, which is also reasonable since the ability to walk indoors or outdoors can be reflected by both movement-related difficulty and the ability/willingness to join social activities. Finally, the revised alternative model also provided evidence that the level of psychological anxiety about losing one's ability to walk in the near future (Q25) are explained by both body pain and cognition. These findings also suggest that the questions might need to be modified or improved to help develop a more parsimonious questionnaire that would still be able to retain the structure behind the definition of LS and the ability to identify affected patients in daily clinical practice. The complex relationships revealed by our CFA between the latent factors and the observed items also provided evidence that current use (single-factor model) of the GLFS-25 was not appropriate or insufficient to capture the underlying clinical status of patients with locomotive organ impairments. Clinicians simply/blindly adding a self-reported rating of unlikely to participate in sports (Q21), meet with friends (Q22), or join in local events (Q23) would end up with a total score of 12, which might lead to overdiagnosis of LS for people who might not be interested in or good at playing sports games and/or attending social events.

We found and proposed a shortened version of GLFS with 16 questions, which not only largely simplified the latent structure by removing the second-order factor, but also reduced most of the cross-loading as well as residual correlations. Moreover, the model fit indices suggested that this may be the best model among all examined models. However, in this shortened GLFS-16 we still had to allow for residual correlations among several indicators and one cross-loading for Q6 on both body pain and movement-related difficulty. The cross-loading of Q6 was reasonable as pain in the body and movement-related difficulty might both account for the daily living matter of sitting down and standing up. The existence of residual correlations suggested that there is non-random measurement error between those correlated indicators. Legitimate reasons for errors of indicators to be related include acquiescent response (response bias caused by the responders agreeing with attitude statements regardless of the content of the question), assessment methods (observer ratings), similarly worded test questions, personal traits (reading disability or cognitive biases such as group think, which affect a respondent's ability to answer a questionnaire truthfully) (*Brown, 2014*). We provided a shortened version of GLFS here, which should not be considered as a definitive guideline for clinical practice but only as a first step for future revision and improvement of the scale.

It is also noteworthy that clinicians may better avoid using the points obtained from responders as if the questions are independently measuring LS. So far, the GLFS is still probably the best available tool to evaluate LS. Clinicians are suggested that when using the scale in evaluating their patients, some questions are more related with each other than the other questions. A shorter version of the scale would possibly be helpful to reduce the time and burden of the responders and avoid confusing/similar wordings which could lead to bias in their answers. However, our shortened GLFS-16 would need to be tested in other samples, future revisions of the scoring algorithm as well as the items that should be included in the questionnaire is still warranted.

Some limitations are worth considering in the current study. First, the participants were chosen via hospital-based recruitment rather than community-based sampling, and the subjects are in pathological condition. Therefore, the generalizability of the findings might be limited and sampling bias may exist. Second, we did not collect the data regarding the other comorbidities. Older patients commonly have multiple comorbidities that may be related to the severity of the physical/psychological latent factors within LS. Third, the concurrent validity of other questionnaires (for assessing quality of life, and other variables) was not assessed in the current study, despite the fact that pain severity, movement-related difficulty, and psycho-social complication were closely associated with LS as shown in previous studies. This concurrency is suggested to be addressed in future studies. Fourth, although the accuracy of participants' answers was not assessed in the current study, nurses and physicians guided the questionnaire completion, which helped us achieved a high response rate and high answer quality but may also be responsible for residual correlations. Finally, although our data showed a Cronbach's $\alpha$ that is as good as that of the developers' (=0.961) (*Seichi et al., 2012*), it should be emphasized that the calculation of Cronbach's $\alpha$ requires the essential assumption of a tau-equivalent model (*Dunn, Baguley & Brunsden, 2014*). In other words, when using the Cronbach's $\alpha$, assuming that the true score variance is constant across all items is unrealistic and seldom achieved in practice (*Green & Yang, 2009*). Therefore, especially when it comes to multi-factor structures, the interpretation of the result of Cronbach's $\alpha$ should be performed cautiously.

## CONCLUSION

In conclusion, the findings of our CFA provided evidence supporting a 3-domain structure of the GLFS in our sample. We proposed three domains of LS (pain in the body, movement-related difficulty, and psycho-social complication) in patients with locomotive disorders. The shortened GLFS may potentially help enhance our understanding of LS from multiple perspectives. Further revision of the scale to clearly discriminate these three constructs is needed.

## ACKNOWLEDGEMENTS

The authors want to express their gratitude to the participants for providing their contributions in joining this study.

### Funding

The publication fee is supported by Aichi Medical University. The funders had no role in study design, data collection and analysis, decision to publish, or preparation of the manuscript.

### Grant Disclosures

The following grant information was disclosed by the authors:
Aichi Medical University.

### Competing Interests

The authors declare that they have no competing interests.

### Author Contributions

- Chaochen Wang conceived and designed the experiments, analyzed the data, prepared figures and/or tables, authored or reviewed drafts of the paper, and approved the final draft.
- Tatsunori Ikemoto conceived and designed the experiments, performed the experiments, analyzed the data, authored or reviewed drafts of the paper, and approved the final draft.
- Atsuhiko Hirasawa performed the experiments, authored or reviewed drafts of the paper, and approved the final draft.
- Young-Chang Arai performed the experiments, authored or reviewed drafts of the paper, and approved the final draft.
- Shogo Kikuchi conceived and designed the experiments, authored or reviewed drafts of the paper, and approved the final draft.
- Masataka Deie conceived and designed the experiments, authored or reviewed drafts of the paper, and approved the final draft.

### Human Ethics

The following information was supplied relating to ethical approvals (i.e., approving body and any reference numbers):

Aichi Medical University Hospital Research Ethics Board approved the study (2019-H136).

### Data Availability

The data, analysis script, figures data are available at GitHub:

https://github.com/winterwang/CFA-GLFS-locomo. The raw data is also available in the Supplemental Files.

### Supplemental Information

Supplemental information for this article can be found online at http://dx.doi.org/10.7717/peerj.9026#supplemental-information.

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
