# Peer review of "Assessment of locomotive syndrome among older individuals: a confirmatory factor analysis of the 25-question Geriatric Locomotive Function Scale"

_PeerJ, doi:10.7717/peerj.9026_

## Round 0.1 · original submission · Major Revisions

Reviewers 2 and 3 made some important comments regarding the statistical analyses conducted by the authors that should be addressed. Reviewers also provided very useful and specific comments that I think they can be useful to improve the quality of the original submision.

Reviewer 1 ·

Basic reporting

The manuscript is written in a clear and concise English that allows the correct interpretation of the data.
In relation to the bibliographical references these are according to the studied topic, updated and support the basic idea of the investigation.
I suggest to the authors that Table 1 be developed in the section on material and methods and not in the introduction.
the theoretical framework proposed is in accordance with the objectives and subsequent results expressed in the research

Experimental design

The article is original and is related to the editorial line of the journal. On the other hand, the research design is clear and can be replicated in other types of populations.

Validity of the findings

The sample size used in this study is a representative sample of the population that allows to interpret the results in a clear way, has a great clinical and replicable applicability.

Additional comments

Dear Authors:

I consider that the work done presents a great clinical applicability, I suggest to remove table 1 of the introduction and incorporate it in the section of material and methods.
The results are in accordance with the stated objectives.

Reviewer 2 ·

Basic reporting

1) The last paragraph of the introduction is a bit hard to follow from a flow perspective. The potential problem of scoring and scale applicability for some individuals (i.e., lines 58-64) is a valid and important point, but it feels out of place in the introduction. Unless this contributes to establishing the problem and/or gap in the literature, it feels like this may fit better in the discussion. For example, does it help set up the problem/gap that this paper will address? The problems that are mentioned are valid, but this paper doesn’t specifically address those problems, so it may be better-suited for the discussion leading into future research.

2) There is a brief mention of factor structure in the introduction (e.g., Tanabe et al., 2018), but this is not elaborated on until the discussion. I think elaborating on the findings of previous literature in this area will help set up the problem and purpose statements more effectively.

3) Models were a bit difficult to follow due to arrows crossing different directions and lack of uniform size (e.g., latent variables different sizes, etc.). It also appears that models present standardized values. If so, some terminology may need to be updated (e.g., error covariances should be error correlation).

4) There are some typos and poor sentence structure in the abstract (e.g., line 52; 32-34), making it difficult to follow what is being conveyed. There were also a couple typos throughout the paper that should be addressed (e.g., line 95). The scale name is also changed in the discussion to GLSF-25 rather than GLFS-25. Sentences on lines 84-86 are unclear with how they are currently written. Some times verb tense seems to switch back and forth (e.g., Methods, lines 124-128).

Experimental design

1) The research question seems to be asking one thing, while the model structure is assessing something slightly different. Specifically, it is a bit confusing as to why sex and age are included in the models depicted in Figures 3 and 4. It seems like the research question is only asking about factor structure of the scale. This is a great first step, especially because this has not been done before. However, because the factor structure has not been confirmed, the relationship with other variables should not be tested yet. If this was a secondary purpose, it needs to be clarified in the paper, but it does not seem to fit as a next step within the scope of this paper and the current findings. I think the goal of identifying the factor structure is adequate given the current results. If sex and age were included for different reasons, then this also needs to be clarified in the paper.

2) Is the six-factor model that is tested in the paper the same as was found in previous literature? This seemed to be the case but was not completely clear with the way that the methods are written currently. Was a single-factor model supported by the literature or was this chosen some other way? A 3-factor structure as identified by Tanabe et al. (2018) through exploratory factor analysis, was also mentioned. Is there a reason this model was not tested in this study using this set of data? This seems like a potentially missed opportunity. These results also need to be compared more extensively in the discussion. Both the present study and Tanabe et al. (2018) found three-factor structures, but were these the same? What are the implications of this?

3) The alternate model generation process seemed to involve mostly the assessment of modification indices. What were the cut-off values used for these decisions? You also mentioned the need to potentially identify a more parsimonious GLFS-25 (line 220). When considering alternate model generation, did you consider the removal of items in an attempt to identify a more parsimonious model?

4) It sounds like the error correlations in the alternate model were statistically supported, but were they also theoretically supported? The theory aspect behind decision-making did not feel like it was addressed or discussed.

5) I think there needs to be a bit more detail in the methods in general.
- For example, how was data cleaned? Were there any missing cases, and if so, how were these addressed?
- The discussion of a 100% response rate, while good information, may be a bit misleading. Instead, how specifically were participants recruited? This detail seemed to be lacking a bit and could provide context about the data collection process.
- What specific cut-off values for CFI and TLI were used?

Validity of the findings

1) Part of the discussion highlights the cross-loadings and the potential issues here. However, the conclusion states that the scale could be useful. I think this needs to be stated with extreme caution, and the limitations and potential issues with the scale need to be addressed further instead, as well as the clinical implications. As a clinician, the most important question is, should we use this scale? If so, how? If not, why not? While the factor structure tells us these things indirectly from a statistical standpoint, it feels like the practical application piece can and should be addressed in more detail.

Reviewer 3 ·

Basic reporting

General Comments: Minor grammatical errors throughout.

Specific Feedback
INTRODUCTION
Well written and provides ample support for why the factor structure of the GLSF-25 should be investigated.

FIGURES
Figure 2 is not necessary; it can be deleted.

Experimental design

General Comments: Research question is defined however, there are significant issues with the methods described.

Specific Feedback
METHODS
Participants
Line 79: error – comma after the semicolon – should be deleted

Model Specification
Line 95: error – comma after the ‘m’ in movement-related
Lines 97-101: consider rephrasing as it is difficult to follow (i.e., difficulty using the stairs, walking briskly, walking without rest, leaving the vicinity, carrying a 2-kg item, doing light level housework, and performing somewhat heavier housework).
Lines 106-107: It is not clear why age and sex of the subjects were treated as observed indicator variables for prediction. If the goal is to establish the factor structure of the GLSF-25 then these should not be included in the CFA. After the factor structure has been identified, invariance testing has been completed (to determine the scale is invariant between sex, age groups, etc.), then a predictive model can be tested. Age and sex should not be included in the CFA models; models must be rerun without and fit-indices reported.

Input data, model estimation, and potential model modification
Line 114-115: How were skewness and kurtosis calculated? What criteria was used? Please include a citation.
Line 118-119: Same as the previous comment
Line 124: CFI and TLI cutoffs are typically reflected as >.90. (previous standard) or >.95 (most recent standard); not ‘close to 1’.
Line 124-125: Although modification indices can be used as a guide, they should not be used alone. MI’s should be assessed but only applied/used if the relationship between the concepts makes theoretical sense.

Validity of the findings

General Comments: There are flaws that need to be addressed regarding analysis and findings. The results, discussion, and conclusions need to be rewritten after the analysis is rerun.

Specific Feedback:
RESULTS AND DISCUSSION
Results
Line 144-146: Cronbach’s alpha should not be calculated for all of the items in the questionnaire. There should be six alphas reported, one for each of the factors. If authors would like to include a Cronbach’s alpha for all of the items because they ran a model that included all items in one factor, it is appropriate however, there should be six alphas reported for each of latent variables. Additionally, the range for alpha should be between 0.7 - 0.9; an alpha above .9 is an indication that multicollinearity may be present.
Lines 154-156: It is not clear why the authors chose to add cross-loadings and residual covariances between latent variables rather than eliminating items. Assessing factor items and content for those items, along with Cronbach’s alpha for each factor would allow you to see the best items to keep/eliminate. Please elaborate on why these steps were taken instead of attempting to identify a more parsimonious model.
Lines 157-158: Although the alternate model has improved fit, it only slightly meets recommended guidelines (CFI and TLI >.95; RMSEA < .06).
Lines 159-166: Alpha should be run on the new factors using all items in the proposed factor.
Line 162: error – should read “as they showed high intercorrelation”
Line 165: error – delete ‘the’; should read “in light of” not “in the light of”
Line 166: error – delete ‘of’ and ‘also’ to read “the factor body pain was positively associated with Q25.
Line 167: error – delete semicolon (;) and start a new sentence with ‘Additionally, Q7…’
Lines 172-176: Although the number of factors was reduced from six to three, there are still issues with the model. Adding residual covariances is only appropriate when it makes sense theoretically. Before doing so, the researchers must look at each of the questions (e.g., are they double-barreled, do they make sense, etc.) and also run the appropriate alphas on the factors. The proposed alternative model included residual covariances that should not be correlated. The model also includes a substantial number of items with low loadings (<.4) and one item that is has a negative valence (when all of the other items are positive). The loadings from second-order factor ‘psychosocial’ to first-order factors ‘soc-daily act’ and ‘cog’ are very high, which is an indication of either multicollinearity or model misspecification.
Authors should seek to condense items and factors to make a parsimonious model. The use of modification indices to only find a statistically better model is inappropriate. Additionally, cross-loadings should not be allowed as this indicates items are not unique to constructs. The alternative model proposed (Figure 4) is rejected as is.
It is highly recommended that authors identify a more parsimonious model (with a reduced number of items) that: 1) theoretically make sense, 2) is psychometrically sound (meets the most up to date criteria), and 3) can be administered and scored with ease. The alternative model proposed does not meet those guidelines.

Discussion
Line 190-192: The sentence that starts “Although dimension” is not clear. I am not sure what the authors are trying to say. Furthermore, dimension and factor analysis are used to identify unique constructs, this might be worth mentioning.
Line 199-203: The goal of the dimension reduction should be to theoretically and statistically differentiate unique latent variables. Even when dimensions are related, the latent variables should be uniquely contributing to the variance of the higher-order construct but in this case, it is not. Constructs can be related but should also be unique. Additionally, the model proposed is confusing for clinicians to use and should therefore not be recommended for practice.
Line 219-222: Completely agree that the questions need to be modified or improved however, it is still possible to create a parsimonious model from the 25-questions. A review of the items should be used as a guide for removal.
Line 224-225: Although you could revise the scoring algorithm, it would be more advantageous to instead design a questionnaire that has three factors with items that uniquely contribute to the factors. The 25-model should be reduced.

CONCLUSION
The results do not support the 3-factor structure and the authors should also address the lack of feasibility for physicians to use the proposed alternative model. The alternative model proposed should not be recommended for use.

---

## Round 0.2 · Minor Revisions

One reviewer still requires several changes to be performed before the manuscript can be considered for publication. Authors should take a careful look at the reviewer´s comment and implement the requested changes in the revised version of the manuscript.

Reviewer 1 ·

Basic reporting

The authors have corrected the requested comments.

Experimental design

The authors have corrected the requested comments.

Validity of the findings

The authors have corrected the requested comments.

Additional comments

I consider that the work is done presents great clinical applicability.

Reviewer 2 ·

Basic reporting

1) The introduction is more concise, but I think it could be set up a bit more effectively. The transition to the problem and purpose statement starting in line 84 is an abrupt transition. I might consider discussing previous research in a bit more depth (e.g., Tanabe et al. study) and then using that to set up the problem, rather than identifying the problem and then discussing the literature that exists.

2) Sentence starting with “The cutoff scores” on line 52 is a bit unclear with how it is currently phrased.

Experimental design

1) Recruitment is described in more detail now, but the sentences describing this process have unclear phrasing. There are also a couple of typos:
a) Line 97 and 99: “eligibility” should be “eligible”
b) Line 98: “GLSF-25” should be “GLFS-25”
c) Line 102: “analysess”

2) The model specification section is a bit confusing with all of the questions listed – is there a more concise way to present this information? Table 1 is helpful in clarifying this information. Is there a way to denote within this table which construct each question represents according to the original scale?

3) I am still a bit confused on whether or not theory was a contributing factor in making modifications to the specified model. It is discussed a bit in the methods section, but when more detail is given later in the manuscript, it feels like there is still a very heavy emphasis on the statistics and numbers in isolation. It is okay to rely on the numbers, but other considerations with theory need to be made. If theory was a contributing factor, I think it needs to be explained a bit more explicitly. If not, I think it needs to be considered when re-specifying the model. If we only use statistics and cut-off values to determine the changes that need to be made, we are likely to find a model that fits regardless. The question is really whether or not that model makes sense theoretically while having good model fit.
a) Related to this, was there a theoretical basis for adding the second-order factor shown in Figure 4?
b) There is one item with a loading of 0.22 under "Body Pain". Is there a reason this was kept? I am curious just because the cut-off given was 0.60.

5) A few typos throughout:
a) Line 197: remove “factor”
b) Line 198 remove “the”
c) Line 201-203: Incomplete sentence

Validity of the findings

Generally, I would be careful of how conclusions are stated.

1) Line 179-181: The interpretation of Cronbach's alpha is a bit confusing as it is currently worded. Currently, it sounds like it is being interpreted as reliability with previous data, instead of the internal consistency within the scale for this present data set, and that it happens to be consistent with previous findings.

2) From a statistical standpoint, the model met cut-off values, so "accepting" it from this standpoint could be okay. But from a theory standpoint, do they hold up? If theory is used, would we end up with this model? (See comments in methods section)

3) It is mentioned that the reliability of the model was confirmed. I'm not sure this can be stated since the final factor structure was different than the original model. There was still a model with 3 factors, but the content of those factors was not consistent with what was originally proposed in the literature. The potential issues of the model are discussed more clearly now, but it feels like sometimes two contrasting ideas are being presented, which makes it a bit hard to follow.

4) Line 275-6: Yes, it could be beneficial to differentiate between these constructs, but the results do not necessarily let clinicians do this with the scale in its current form. I would just be careful in the way this information is presented, as it sounds like you are saying that the current scale with 3 constructs can help do this. As you mention, there are cross-loadings, etc. within the final model, but these make it difficult to differentiate between constructs.

5) Line 293-298: Valid points, but some of this information seems out of place. I would make sure to set up this point in your discussion more clearly.

6) Good discussion points overall in the paragraph on lines 300-320. A couple things to consider:
a) Sentence starting on Line 313 with “Our analyses…” – I would be careful with this statement – Unless compared to other tools, this claim may be difficult to make.

7) Fourth limitation – this doesn’t necessarily feel like a limitations methodologically as much as it does a discussion consideration.

8) I might consider modifying the conclusion slightly in accordance with any other changes made.

A few typos throughout:
a) Line 247-250: Incomplete sentence; needs revising.
b) Line 255: "explanatory" should be replaced with "exploratory"
b) Line 262: should be “3 different” instead of “different 3”
c) Line 319-20: Comma splice

---

## Round 0.3 · Minor Revisions

Please, consider the remaining comments of the reviewer.

Reviewer 2 ·

Basic reporting

Abstract:
Line 19: “have not” should be “has not”


Introduction:
Line 60: Sentence starting “The full version…” is a bit confusing as it is currently worded.


Methods:
Line 78-79: “500 samples in order of visiting” – phrasing is confusing here. Is this intending to say that questionnaires were distributed until a total of 500 had been completed and collected?

Line 94: “written consent form of the study” – phrasing is confusing.

Lines 122 and 124: Remove bold period after table numbers to keep consistent with rest of manuscript

Line 131-132: “CFI and TLI are both required to be…” – this phrasing is a bit hard to follow with the rest of the sentence structure. Could potentially just put (≥0.95) after CFI and TLI as with the previous fit indices.

Line 134: “is” should be “was”

Line 138: “are” should be “were”

Line 139: “will” should be “would”; “fitting” should be “fit”


Results:
General consideration – Be sure that factor names, capitalization, etc. is consistent throughout.

Line 154: Does “Their mean age” refer to only the females or the full sample? It is not clear given the previous sentence.

Line 165: “fitting” should be “fit”

Line 185: “on factor of…and factor of…” – phrasing is unclear.

Line 186: “fitting” should be “fit”

Lines 200-201: This sentence is unclear. Is this saying that Q6 loaded onto two factors?


Discussion:
Line 213: comma splice; consider starting a new sentence after “with aging”.

Line 224: “fitting” should be “fit”

Line 238: remove “the” before “psycho-social complications” since “and LS” is also listed afterwards.

Line 242: unclear wording – either needs “the” added before “full 25 questions” or could also say “all 25 questions” instead.

Line 280: “are suggested” is unclear here. Perhaps something like “should note that…”

Line 284-5: Wording is unclear – is this meant to say “as well as the items [that] should be included…”?

Line 287: “subject” should be “subjects”

Line 296: “answers quality” should be “answer quality”

Line 297: “maybe also responsible” should be “may also be responsible”


Conclusion:
Line 308: is the word “organ” supposed to be included?

Line 309: “enhancing” should either be “in enhancing” or just “enhance”

Experimental design

No comment

Validity of the findings

No comment

Additional comments

The authors have made positive improvements to this manuscript.

---

## Round 0.4 · accepted · Accept

Congratulations for meeting the high standard for publishing in PeerJ